# Parameter Estimation of Host Genomic and Gut Microbiota Contribution to Growth and Feed Efficiency Traits in Meat Rabbits

**DOI:** 10.3390/microorganisms12102091

**Published:** 2024-10-19

**Authors:** Xinyang Tian, Junkun Zhou, Yinghe Qin, Kai Zhang, Wenqiang Sun, Song-Jia Lai, Xianbo Jia, Shi-Yi Chen

**Affiliations:** 1Farm Animal Genetic Resources Exploration and Innovation Key Laboratory of Sichuan Province, Sichuan Agricultural University, Chengdu 611130, China; tianxinyang145@gmail.com (X.T.); junkunzhou1@gmail.com (J.Z.); wqsun2021@sicau.edu.cn (W.S.); laisj5794@163.com (S.-J.L.); 2College of Animal Science and Technology, China Agricultural University, Beijing 100193, China; qinyinghe@cau.edu.cn; 3Sichuan Academy of Grassland Sciences, Chengdu 611743, China; zkscgrassland@163.com

**Keywords:** growth, heritability, microbiability, ssGBLUP, MBLUP

## Abstract

Rabbits can efficiently utilize plant fibers that are indigestible to humans, and hence may contribute to the alleviation of feed–food competition. Therefore, it is economically and ecologically important to genetically improve the growth performance and feed efficiency of meat rabbits. In this study, we combined pedigree, genomic, and gut microbiota data to estimate genetic and microbial parameters for nine growth and feed efficiency traits of 739 New Zealand White rabbits, including body weight (BW) at 35 (BW35), 70 (BW70), and 84 (BW84) days of age, and average daily gain (ADG), feed conversion ratio (FCR), and residual feed intake (RFI) within two age intervals of 35–70 days (ADG70, FCR70, and RFI70) and 35–84 days (ADG84, FCR84, and RFI84). Based on single-step genomic best linear unbiased prediction, three BW traits and two ADG traits had the high estimates (±standard error, SE) of heritability, ranging from 0.44 ± 0.13 of BW35 to 0.66 ± 0.08 of BW70. Moderate heritabilities were observed for RFI70 (0.22 ± 0.07) and RFI84 (0.29 ± 0.07), whereas the estimates did not significantly deviate from zero for the two FCR traits. There was moderate positive genetic correlation (±SE) between BW70 and ADG70 (0.579 ± 0.086), but BW70 did not correlate with RFI70. Based on microbial best linear unbiased prediction, the estimates of microbiability did not significantly deviate from zero for any trait. Based on the combined use of genomic and gut microbiota data, the parameters obtained in this study could help us to implement efficient breeding schemes in meat rabbits.

## 1. Introduction

Rabbits (*Oryctolagus cuniculus*) are one of the most recently domesticated livestock [1], and have been popularly raised in Asian and European countries for producing meat. China ranked the highest in production of rabbit meat with 358 thousand tons in 2022 [2]. Among the common meat sources of livestock, rabbit meat is one of the healthiest white meats [3]. As a prolific small herbivorous livestock, rabbits can efficiently utilize plant fiber fractions that are indigestible to humans. Therefore, raising rabbits is expected to greatly contribute to the alleviation of feed–food competition [4]. In this context, it may be economically and ecologically important to improve the growth performance and feed efficiency of meat rabbits, especially using genetic selection approaches.

In meat rabbits, individual growth performance is always measured as live weight at various ages, as well as the derived daily gains. On the whole, these growth-related traits are moderately heritable in meat rabbits, which means that they can be genetically improved efficiently [5]. Besides growth performance, feed efficiency in meat rabbits has been gaining more and more attention during the past years [6]. However, it is highly time-consuming and expensive to collect individual feed efficiency phenotype in a large-scale population study. Genomic information has been widely used into best linear unbiased prediction (BLUP) and Bayesian models regarding genetic evaluation of economically important, hard-to-measure, and lowly heritable traits in livestock [7]. In comparison with other common livestock, however, genomic evaluation in rabbits has not been effectively implemented yet. Using the simulated genotype, Mancin et al. [8] first proposed that genomic selection is feasible in rabbits. Recently, low-coverage genome sequencing data were also used for estimating genomic parameters of wool traits in Angora rabbits [9].

Besides host genetic background, gut microbiota composition may have significant impacts on individual growth and feed efficiency in rabbits [10]; and the underlying biological mechanisms referred to are mainly gut microbiota-mediated modifications on nutrient utilization efficiency of diets and host health conditions [11]. In ruminants, gastrointestinal microbiota have been extensively involved in the digestion of recalcitrant dietary fiber [12], which can further impact host growth performance and feed efficiency [13,14]. However, the impacts of gut microbiota on the digestion of dietary fiber, growth, and feed efficiency have not been thoroughly explored yet in rabbits [10,15]. Similar to the genomic evaluation, two common methods have been used for dissecting the contribution of microbiota to individual phenotype variation, including the association analysis, and partitioning of phenotypic variance components. Microbiome-wide association studies facilitate the discovery of specific microbial species that significantly influence the phenotype of interest [16]. In contrast, partitioning of phenotypic variance attributed to microbiota composition variation could help us explore their overall contribution to complex traits, which was first proposed in Holstein cows and termed as microbiability by Difford et al. [17]. In other words, microbiability is the proportion of phenotypic variance that can be explained by the microbiota composition variation among individuals. Obviously, the estimation of microbiability is analogous to that of heritability in terms of theory and methodology. Due to the technological advances and biologic implications, many studies have been carried out during recent years on combining genomic with microbiota data for the genetic evaluation of economically important and hard-to-measure traits [18], such as methane emission traits in dairy cows [19], fat deposition traits in chickens [20], meat quality and carcass composition traits in swine [21], feed efficiency traits in Holstein cows [22], and milk composition traits in sheep [23]. The combined use of genomic and gut microbiota data may improve estimation accuracy of the related genetic parameters and individual breeding values, and also help to reveal the possible interactions between host genetic background and gut microbiota [18,24].

In addition to the impacts of environmental factors, such as diets and living conditions, on gut microbiota composition [25], it has been found that the relative abundance of some microbial species could be controlled by host genetic background and herein considered heritable [26,27]. In this context, Weishaar et al. [28] first proposed a two-step approach to decompose individual breeding value into two parts, including direct and microbiota-mediated indirect contributions of host genes on phenotype. Subsequently, Christensen et al. [29] developed a joint model that can effectively combine two or multiple omics data into a single-step model. Recently, multi-omics data-based genetic evaluation has been increasingly reported in livestock, especially to combine genomic and gut microbiota data for evaluating feed efficiency traits in chickens [30], pigs [31], and Holstein cows [22]. Under the hypothesis that host genetic and/or gut microbiota impact the growth and feed efficiency in rabbits, therefore, the main objectives of this study are to (1) measure and compare different growth and feed efficiency traits in meat rabbits for exploring their phenotypic correlations, and (2) combine host genomic and gut microbiota data for estimating their contributions to these traits and obtaining the related parameters.

## 2. Materials and Methods

### 2.1. Animals and Phenotype

Phenotypic data were initially collected among 739 New Zealand White rabbits; all of them were the offspring of 101 multiparous females randomly mated with 72 males. New Zealand White rabbits were selected as that is an excellent breed of meat rabbits. We did not have the upward pedigree information regarding these founder individuals that were randomly selected from a purebred population raised in our research farm. All rabbits were born in two different batches using the lighting-assisted estrus synchronization technology [32]. Between the weaning at 35 days of age and finishing at 84 days of age, each rabbit was individually housed per cage and fed ad libitum with either of two commercial pellet diets (labeled as digestible energy = 10.5 MJ, protein = 15.5%, and crude fiber = 16.5%) that only differed in the source of mineral supplements. The use of different mineral supplements was intended to compare the effects on individual growth performances in our other study. The air conditioning control system was used when indoor temperature was higher than 25 °C. Other management practices had been kept consistent throughout the experimental period.

All rabbits were euthanized using an overdose of anesthesia at 84 days of age. Before the morning feeding, individual body weight (BW) was measured using same electronic scale (to an accuracy of 1 g) at 35 (BW35), 70 (BW70), and 84 (BW84) days of age, respectively. The individual average daily gain (ADG) was calculated for the two age intervals of 35–70 days (ADG70) and 35–84 days (ADG84), respectively. Furthermore, the feed intake of each rabbit was individually collected at 70 and 84 days of age, and feed conversion ratio (FCR) was therefore derived for the two age intervals as FCR70 (i.e., between 35 and 70 days) and FCR84 (i.e., between 35 and 84 days). Similarly, residual feed intake (RFI, g/day) was further calculated for the two age intervals (RFI70 and RFI84) as ${FI}_{i}=\mu +\alpha \times {{midBW}_{i}}^{0.75}+\beta \times {ADG}_{i}+{RFI}_{i}$, where ${FI}_{i}$ is the average feed intake (g/day) and ${{midBW}_{i}}^{0.75}$ is mid-metabolic BW (g) for animal i; μ, α, and β are the regression coefficients of mean, metabolic BW, and ADG, respectively [33]. For each of these phenotype traits, the possible outliers were removed if they were outside the median ± 3.5 × median absolute deviation [34].

### 2.2. Host Genotyping and Profiling of Gut Microbiota

At the finishing day, blood samples were collected from 199 rabbits roughly consisting of one male and one female randomly selected per litter. One μg genomic DNA of each rabbit was used for high-throughput sequencing of 50 K targeted genomic loci provided by Compass Biotechnology Inc. (Beijing, China). Raw reads were quality controlled using fastp software v0.23.3 [35] and aligned against the reference genome using BWA software v0.7.17 [36]; a total of 48,916 raw single nucleotide polymorphisms (SNPs) were initially obtained using GATK software v4.0 [37]. SNPs were further removed if they had the calling rate < 0.9, minor allele frequency < 0.05, or extreme deviation from the Hardy–Weinberg equilibrium (*p* < 10^−8^) using plink software v1.9 [38]. Missing genotypes were further imputed using Beagle software v5.3 [39], after which a total of 41,359 SNPs remained for the 199 rabbits.

At 49 days of age, rectal feces samples were successfully collected from 707 rabbits. Bacterial genomic DNA was extracted using a QIAamp BiOstic Bacteremia DNA Kit (Qiagen, Shanghai, China) and subjected to amplification of the V3–V4 region of the 16S rRNA gene using the universal primers and HOTSTAR Taq Plus Master Mix Kit (Qiagen, Shanghai, China). Libraries were sequenced on the Illumina HiSeq™ 2000 platform for generating 300 bp paired-end reads. Bioinformatic analyses of sequencing reads were conducted using QIIME2 v2024.2 [40]. In brief, the paired reads were merged if they overlapped at least 20 bp in length and no more than 3 bp mismatches using VSEARCH software v2.26.1 [41]. Low-quality sequences were removed by the sliding window (5 bp) with an average Qscore < 30, and all sequences were then trimmed to 400 bp after discarding the first 20 nucleotides at the 5′ end. After discarding artificial and chimera sequences using Deblur software v1.1.1 [42], amplicon sequence variants (ASVs) were produced and randomly rarefied to 17,688 sequences to address different sequencing depths across samples. The low-abundance of ASVs that were present in less than 20% of samples were discarded and the constant value of 1 was added to every element for handling the zero values.

### 2.3. Statistical Models of Single Traits

There were three categorical fixed effects recorded, including the gender (two levels), birth batch (two levels), and diet group (two levels), which herein were selected for inclusion into the models based on the backward elimination procedure (*p* < 0.05) of the *lm* function in the R software v4.3.3 [43], separately for each trait. Based on the Akaike Information Criterion (AIC) and likelihood-ratio testing, we also compared whether the random maternal permanent environmental effect needs to be further included into the models or not.

For each trait, genetic analyses were first performed using three statistical models, including the single-step genomic BLUP (G), microbial BLUP (M), and single-step genomic plus microbial BLUP (GM). The model G was defined as:$$y=Xb+Wp+Z_{1}a+e\;\;\;\;(G)$$where y is the vector of phenotypic records; b is the fixed-effect vector described in Table 1; p is the vector of maternal permanent environmental effects if included for the trait analyzed, with $p\sim N(0,I\sigma_{p}^{2})$; a is the vector of animal additive genetic effects, with $a\sim N\left(0,H\sigma_{a}^{2}\right)$; e is the vector of random residuals, with $e\sim N(0,I\sigma_{e}^{2})$. X, W, and $Z_{1}$ are the design matrix for b, p, and a, respectively. I is an identity matrix. H is a hybrid relationship matrix that combines the genomic with pedigree information [44], and the inverse of H used can be computed as follows [45]:$$H^{-1}=A^{-1}+\left[\begin{array}{cc} 0 & 0\\ 0 & G^{-1}-A_{22}^{-1} \end{array}\right],$$
where $A^{-1}$ is the inverse of the pedigree relationship matrix; $A_{22}^{-1}$ is the inverse of the pedigree relationship matrix for these genotyped animals; and $G^{-1}$ is the inverse of the genomic relationship matrix (G) that was derived using method of VanRaden [46]:$$G=\frac{ZZ^{\prime }}{2\sum p_{i}(1-p_{i})},$$
where Z is the matrix of genotypes adjusted for the allele frequencies; $Z^{\prime }$ is the transpose of Z; and $p_{i}$ is the allele frequency of marker i.

Using gut microbiota information, the model M could be similarly defined as follows:$$y=Xb+Wp+Z_{2}m+e,\;\;\;\;(M)$$where m is the vector of animal gut microbiota effects, with its design matrix of $Z_{2}$. The assumed distribution is $m\sim N\left(0,O\sigma_{m}^{2}\right)$, and $\sigma_{m}^{2}$ is variance of gut microbiota effects. O is the microbial relationship matrix with a n×n dimension, and was constructed as $O=MM^{\prime }/m$ according to the method of Rose et al. [47]. Here, n and m are the number of animals and ASVs, respectively; M is the log-transformed and normalized abundance matrix, with its element of $m_{jk}$ computed as follows:$$m_{jk}=\frac{logP_{jk}-mean(\log P_{.k})}{sd(\log P_{.k})},$$
in which $P_{jk}$ is the relative abundance of ASV k observed in animal j; $P_{.k}$ is the vector containing the relative abundance of ASV k among the n animals. The other terms were defined above.

When further considering host genomic and gut microbiota information together, the model GM was defined as follows:$$y=Xb+Wp+Z_{1}a+Z_{2}m+e,\;\;\;\;(\mathrm{GM}).$$

Here, these terms were defined above.

### 2.4. Bivariate Models

For these heritable traits, pairwise genetic correlations were further estimated based on bivariate analyses of model G, that were defined as follows:$$\left[\begin{array}{c} y_{1}\\ y_{2} \end{array}\right]=\left[\begin{array}{cc} X_{1} & 0\\ 0 & X_{2} \end{array}\right]\left[\begin{array}{c} b_{1}\\ b_{2} \end{array}\right]+\left[\begin{array}{cc} W_{1} & 0\\ 0 & W_{2} \end{array}\right]\left[\begin{array}{c} p_{1}\\ p_{2} \end{array}\right]+\left[\begin{array}{cc} Z_{1_{1}} & 0\\ 0 & Z_{1_{2}} \end{array}\right]\left[\begin{array}{c} a_{1}\\ a_{2} \end{array}\right]+\left[\begin{array}{c} e_{1}\\ e_{2} \end{array}\right],$$
where $y_{1}$ and $y_{2}$ are vectors of phenotypic records for the two traits analyzed, together with their fixed effects ($b_{1}$ and $b_{2}$), (if included) maternal permanent environmental effects ($p_{1}$ and $p_{2}$), additive genetic effects ($a_{1}$ and $a_{2}$), and random residuals ($e_{1}$ and $e_{2}$). The assumptions of additive genetic, maternal environmental, and residual effects are as follows:$$\left[\begin{array}{c} a_{1}\\ a_{2} \end{array}\right]\sim N\left(0,H⊗\left[\begin{array}{cc} \sigma_{a_{1}}^{2} & \sigma_{a_{1}a_{2}}\\ \sigma_{a_{1}a_{2}} & \sigma_{a_{2}}^{2} \end{array}\right]\right),$$
$$\left[\begin{array}{c} p_{1}\\ p_{2} \end{array}\right]\sim N\left(0,I⊗\left[\begin{array}{cc} \sigma_{p_{1}}^{2} & \sigma_{p_{1}p_{2}}\\ \sigma_{p_{1}p_{2}} & \sigma_{p_{2}}^{2} \end{array}\right]\right),\mathrm{and}$$
$$\left[\begin{array}{c} e_{1}\\ e_{2} \end{array}\right]\sim N\left(0,I⊗\left[\begin{array}{cc} \sigma_{e_{1}}^{2} & \sigma_{e_{1}e_{2}}\\ \sigma_{e_{1}e_{2}} & \sigma_{e_{2}}^{2} \end{array}\right]\right),\mathrm{respectively}$$where $\sigma_{a_{1}}^{2}$ and $\sigma_{a_{2}}^{2}$ are the additive genetic variances, $\sigma_{p_{1}}^{2}$ and $\sigma_{p_{2}}^{2}$ the maternal environmental effect variances, and $\sigma_{e_{1}}^{2}$ and $\sigma_{e_{2}}^{2}$ the residual variances for the two traits analyzed, respectively; $\sigma_{a_{1}a_{2}}$, $\sigma_{p_{1}p_{2}}$, and $\sigma_{e_{1}e_{2}}$ are their respective covariances between the two traits. The other terms were defined above.

### 2.5. Estimates of Genetic Parameters

Based on the statistical models described above, we accordingly derived the relevant parameters, including the total heritability with $h^{2}=\sigma_{a}^{2}/\left(\sigma_{a}^{2}+\sigma_{p}^{2}+\sigma_{e}^{2}\right)$ derived from model G, total microbiability with $m^{2}=\sigma_{m}^{2}/\left(\sigma_{m}^{2}+\sigma_{p}^{2}+\sigma_{e}^{2}\right)$ from model M, direct heritability with $h_{d}^{2}=\sigma_{a_{d}}^{2}/\left(\sigma_{a_{d}}^{2}+\sigma_{m_{d}}^{2}+\sigma_{p}^{2}+\sigma_{e}^{2}\right)$, and direct microbiability with $m_{d}^{2}=\sigma_{m_{d}}^{2}/\left(\sigma_{a_{d}}^{2}+\sigma_{m_{d}}^{2}+\sigma_{p}^{2}+\sigma_{e}^{2}\right)$ from model GM. According to the bivariate models, genetic correlations between two traits were calculated as $r_{g}=\sigma_{a_{1}a_{2}}/\sqrt{\sigma_{a_{1}}^{2}\sigma_{a_{2}}^{2}}$. All variance components were estimated using the Average-Information Restricted Maximum Likelihood (AI-REML) method implemented in the AIREMLF90 software v202405 [48,49].

## 3. Results

### 3.1. Phenotype and Correlations

Descriptive statistics of the phenotype traits are in Table 1. In total, the used number of records varied from 700 for RFI70 to 729 for BW35. Means (±standard deviation, SD) of BW35, BW70, and BW84 were 843.5 ± 134.4 g, 2058.8 ± 210.6 g, 2504.7 ± 244.2 g, respectively. ADG70 (34.7 ± 4.6 g/day) was greater than ADG84 (33.9 ± 3.8 g/day), whereas FCR70 (3.6 ± 0.4) was lower than FCR84 (4.0 ± 0.4). The estimates of RFI70 and RFI84 were −1.3 ± 10.1 g/day and −1.4 ± 10.2 g/day, respectively. Phenotypic distributions and pairwise correlations are in Figure 1. BW35 had significant (*p* < 0.001) and moderate positive correlations with BW70 (*r* = 0.638), BW84 (*r* = 0.610), FCR70 (*r* = 0.415), and FCR84 (*r* = 0.403), whereas it did not correlate with ADG70, ADG84, RFI70, and RFI84. Among the four measures at 70 days of age, BW70 had a moderate correlation with ADG70 (*r* = 0.769; *p* < 0.001), but weak negative and weak positive correlations with FCR70 (*p* < 0.01) and RFI70 (*p* < 0.05), respectively; FCR70 had moderate correlations with ADG70 (*r* = −0.505; *p* < 0.001) and RFI70 (*r* = 0.573; *p* < 0.001), whereas only a weak correlation was present between ADG70 and RFI70 (*r* = 0.115; *p* < 0.001). Considerable and significant (*p* < 0.001) correlations were observed between BW70 and BW84 (*r* = 0.874), ADG70 and ADG84 (*r* = 0.807), FCR70 and FCR84 (*r* = 0.692), and RFI70 and RFI84 (*r* = 0.685).

### 3.2. Model Comparisons

For the nine traits, the included fixed effects and comparison results of three statistical models are in Table 1 and Table 2. Individual gender had significant effects on BW35, BW70, FCR70, and FCR84, while diet group had significant effects on BW70, ADG70, ADG84, and RFI70. There were significant effects of birth batch on all traits, with an exception of BW35. Based on AIC values and likelihood-ratio testing (*p* < 0.01), the candidate random effect of maternal permanent environments was included for BW35, FCR70, and FCR84 in model G, for all nine traits in model M, and for BW35, FCR70, and FCR84 in model GM, respectively.

### 3.3. Estimates of Heritability, Microbiability, and Genetic Correlations

Based on the best fit model selected for each trait, estimates of heritability and microbiability are in Table 3. For the model G, the three BW traits and two ADG traits had high estimates of heritability (±SE), ranging from 0.44 ± 0.13 of BW35 to 0.66 ± 0.08 of BW70. Moderate heritabilities were estimated for RFI70 (0.22 ± 0.07) and RFI84 (0.29 ± 0.07), whereas the estimates of heritability did not deviate from zero with 0.11 ± 0.12 of FCR70 and 0.07 ± 0.12 of FCR84. Based on model M, there was no detectable microbiability for any of the nine traits, whose estimates ranged from 0.01 ± 0.02 for ADG70 and RFI84 to 0.08 ± 0.04 for ADG84. Therefore, the estimates of heritability based on model GM did not obviously change in comparison with that in model G.

Pairwise genetic correlations are in Table 4 among the seven BW, ADG, and RFI traits that showed the relatively high or moderate heritabilities. BW35 had strong positive genetic correlations (±SE) with BW70 (0.762 ± 0.049) and BW84 (0.708 ± 0.054), whereas it did not show genetic correlations with ADG70 (−0.092 ± −0.095), ADG84 (0.023 ± 0.106), RFI70 (−0.150 ± 0.169), and RFI84 (−0.061 ± 0.134). There were moderate positive genetic correlations between BW70 and ADG70 (0.579 ± 0.086), and BW84 and ADG84 (0.720 ± 0.059), whereas BW70 did not correlate with RFI70 (−0.113 ± 0.197). Furthermore, we observed the nearly complete genetic correlations between BW70 and BW84 (1.000 ± 0.001), ADG70 and ADG84 (1.000 ± 0.002), and RFI70 and RFI84 (0.937 ± 0.076).

## 4. Discussion

Growth performance is economically important in meat rabbits and has been intensively subjected to genetic selection and improvement [5]. Finishing BW and post-weaning ADG are two main types of traits for measuring individual growth of rabbits, as well as for other meat livestock. In China, meat rabbits are usually finished at either 70 or 84 days of age, depending on the breed-specific growth rate and market demands. To better address the challenges of global food security and carbon emissions, attention has been increasingly paid during past years to evaluating and improving feed efficiency traits in various livestock [50,51]. FCR is the most extensively used trait measuring feed efficiency in the literature, which is calculated as the ratio of individual feed intake to weight gain during a specific time interval [52]. Another common trait measuring feed efficiency is the RFI, which properly adjusts individual metabolic BW and ADG [53]. In this study, we compared nine growth and feed efficiency traits in meat rabbits; and the results suggested that these rabbits finished at 70 days of age may have greater ADG and lower FCR and RFI than those finished at 84 days. Liao et al. [54] fitted individual growth curves in a crossbred population of meat rabbits and similarly found the point of growth inflection occurred around 70 days of age. Collectively, these results suggest that this population of New Zealand White rabbits finished at 70 days of age would have higher growth performance and feed efficiency.

During the past decade, genomic BLUP approaches have been widely used for genetic evaluation in various livestock, especially regarding these low heritability and hard-to-measure traits [7,55]; whereas these state-of-the-art approaches have not been effectively implemented in meat rabbits yet. In this study, we combined genomic and pedigree information to perform genetic evaluation on several important growth and feed efficiency traits in meat rabbits. Our results revealed the high estimates of heritability for all BW and ADG traits. The previous estimates of heritability solely based on the pedigree information were 0.443 ± 0.02 and 0.297 ± 0.03 for the BW at 30 and 90 days of age in New Zealand White rabbits, respectively [56]. The heritabilities estimated for BW at weaning and the end of the fattening period were 0.033 ± 0.013 and 0.059 ± 0.020 in a synthetic rabbit line, respectively [57]. El-Deghadi et al. [58] estimated the heritability of 0.24 ± 0.01 for BW84 in an Egyptian rabbit breed. Therefore, our estimates in this study are somewhat higher than previous reports found in the literature, which may be due to the differences in population genomic architecture, environmental condition, and genetic relationship information constructed. In contrast to growth traits, we found that two FCR traits were not heritable in our population, whereas the two RFIs were moderately heritable. Gidenne et al. [6] comprehensively reviewed genetic and non-genetic aspects of feed efficiency in rabbits, and they showed that FCR and RFI traits are moderately heritable. Therefore, our observations in this study would be partially inconsistent with previous reports, which may indicate inter-population variation and the requirement to perform a case-by-case estimation of genetic parameters. A previous study similarly revealed that genetic correlations between feed efficiency and growth traits may be varied between ad libitum and restricted feeding regimens [6]. Based on genome-wide association studies, Sánchez et al. [59] and Garreau et al. [60] found tens to hundreds of SNPs that are significantly associated with FCR and RFI traits in rabbits, whereas both studies did not estimate the heritability. As only a proportion of rabbits (~30%) were genotyped in this study, the increased number of genotyping is expected to increase the estimation accuracy of genetic parameters. Collectively, we suggested that the traits used to genetically improve feed efficiency in rabbits need to be specifically evaluated before including them into the breeding schemes.

Although there were only moderate phenotypic correlations of BW35 with BW70 and BW84 observed in this study, we found greater genetic correlations among them. Our results were consistent with previous estimates on genetic correlation between weaning and fattening BW in New Zealand White rabbits [61]. These results suggest that, therefore, it is possible to select individuals based on the genetic merits evaluated at weaning BW for indirectly improving the finishing BW, while the accuracy may be decreased if we only use their phenotypic information at the weaning age. However, it is less efficient to indirectly select ADG only based on the genetic merits evaluated at the finishing BW as only moderate genetic correlations were present between them, although actually they had the strong phenotypic correlations. In our population, RFI is a promising trait used for genetically improving feed efficiency in meat rabbits as moderate heritability was observed, whereas it is impossible to indirectly improve feed efficiency according to genetic merits evaluated at the finishing BW because no genetic correlation was present between them. For BW, ADG, and RFI, strong phenotypic and nearly complete genetic correlations were observed between 70 and 84 days of age, which means that the two traits could be replaceable with each other into breeding schemes. For example, only either BW70 or BW84 needs to be included for selection. Sakthivel et al. [61] systematically estimated phenotypic and genetic correlations among multiple growth traits in New Zealand White rabbits, and revealed similar results as we observed in this study. However, phenotypic or genetic correlations of feed efficiency traits with these traditional growth traits in meat rabbits have not been commonly found in literature yet. On the whole, both FCR and RFI traits measured in this study showed no or low phenotypic and genetic correlations with different BW and ADG traits in meat rabbits.

Due to potential biological implications of gut microbes on regulating host growth and feed efficiency, as well as the technological advances on microbiota profiling, gut microbiota data have been increasingly used for evaluating feed efficiency and growth traits in various livestock [18,20,21,22,62,63]. Collectively, these studies suggested positive contributions of gut microbiota to the phenotypic variation of traits studied. For example, the estimates of microbiability were suggested to be relatively moderate for feed efficiency traits in pigs [62], dairy cattle [22], and chickens [63]. In rabbits, Velasco-Galilea et al. [10] also reported high estimates of microbiability for the feed efficiency traits and suggested that the inclusion of microbial information could obviously improve the predictability for cage-average feed efficiency and individual growth traits. On the other hand, the effects of continuous selection on environmental variance could significantly shift gut microbiota composition in rabbits [64]. In contrast to these reports, we did not detect meaningful contribution of gut microbiota to either the feed efficiency or growth traits involved in this study, which may be essentially resulted from inter-population differences. However, additional comparisons on different statistical models, long-read sequencing of the 16S rRNA gene, and the development of novel indicators are required in future studies to explore potential correlations of gut microbiota with growth and feed efficiency in rabbits. Anyway, our negative results suggested that there may be variable involvement of gut microbiota in regulating host growth and feed efficiency. Therefore, some caution is required when using gut microbiota data for phenotypic prediction on feed efficiency and growth traits in rabbits, or even as well as other livestock. For every trait analyzed in this study, meanwhile, we found that the maternal permanent environments have significant effects on regressing gut microbiota on the phenotype by linear mixed models. This may coincide with the fact that gut microbiota composition is especially apt to be influenced by common environmental factors, such as living condition and diet [65]. Therefore, our results also suggested that specific attention should be paid to exclude possible confounding environmental factors, which, if not adjusted properly in the statistic models, may result in the overestimation of microbiota contribution to the phenotypic variation of traits of interest. Furthermore, it is well-known that gut microbiota composition would be dynamically changed along with individual growth. In this study, the 49 days of age of rabbits was selected for sampling the feces samples because of two considerations. First, the gut microbial composition may be relatively stable as it was two weeks after the weaning date. Second, at least three weeks remained before the possible finishing date (may be varied from 70 to 84 days of age among different breeds or markets), which could facilitate the obtainment of gut microbiota data for conducting individual selection.

## 5. Conclusions

In this study, we successfully used genomic and gut microbiota data for evaluating their contributions to several important growth and feed efficiency traits in meat rabbits. Our estimates of heritability were relatively high for all growth traits, whereas only one of the two feed efficiency traits showed moderate heritability. These results suggested that we could genetically improve growth performance in meat rabbits, but the trait(s) used to improve feed efficiency should be specially evaluated before including them in the breeding schemes. In contrast to previous reports in the literature, no contribution of gut microbiota to phenotypic variation was observed for any trait analyzed in this study; however, these negative results may be further re-estimated using different statistical models, higher resolution on microbiota composition profiling, as well as an increased sample size.

## Figures and Tables

**Figure 1 microorganisms-12-02091-f001:**
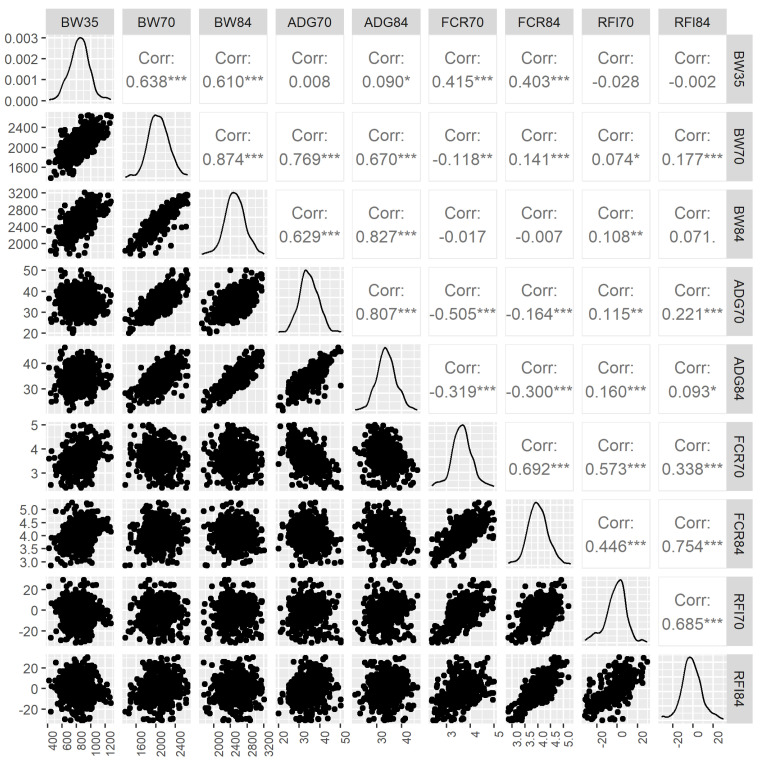
Phenotypic correlations among the growth and feed efficiency traits. BW35, live body weight at 35 days of age (g); BW70, live body weight at 70 days of age (g); BW84, live body weight at 84 days of age (g); ADG70, average daily gain between 35 and 70 days of age (g/day); ADG84, average daily gain between 35 and 84 days of age (g/day); FCR70, feed conversion ratio between 35 and 70 days of age; FCR84, feed conversion ratio between 35 and 84 days of age; RFI70, residual feed intake between 35 and 70 days of age (g/day); RFI84, residual feed intake between 35 and 84 days of age (g/day). (* *p* < 0.05, ** *p* < 0.01, *** *p* < 0.001).

**Table 1 microorganisms-12-02091-t001:** Descriptive statistics for traits and effects included in the mixed models for each trait.

Traits ^a^	Descriptive Statistics ^b^			Fixed Effects ^c^	Random Effects ^d^		
	N	Mean	SD		G	M	GM
BW35	729	843.5	134.4	s	a, p	m, p	a, m, p
BW70	724	2058.8	210.6	s, b, d	a	m, p	a, m
BW84	723	2504.7	244.2	b	a	m, p	a, m
ADG70	725	34.7	4.6	b, d	a	m, p	a, m
ADG84	720	33.9	3.8	b, d	a	m, p	a, m
FCR70	714	3.6	0.4	s, b	a, p	m, p	a, m, p
FCR84	714	4.0	0.4	s, b	a, p	m, p	a, m, p
RFI70	700	−1.3	10.1	b, d	a	m, p	a, m
RFI84	705	−1.4	10.2	b	a	m, p	a, m

^a^ BW35, live body weight at 35 days of age (g); BW70, live body weight at 70 days of age (g); BW84, live body weight at 84 days of age (g); ADG70, average daily gain between 35 and 70 days of age (g/day); ADG84, average daily gain between 35 and 84 days of age (g/day); FCR70, feed conversion ratio between 35 and 70 days of age; FCR84, feed conversion ratio between 35 and 84 days of age; RFI70, residual feed intake between 35 and 70 days of age (g/day); RFI84, residual feed intake between 35 and 84 days of age (g/day). ^b^ N, the number of records; SD, standard deviation. ^c^ s, gender; b, birth batch; d, diet group. ^d^ G, model G; M, model M; GM, model GM; a, animal additive genetic effect; p, maternal permanent environmental effect; m, animal microbiota effect.

**Table 2 microorganisms-12-02091-t002:** Comparisons to include maternal permanent environmental effect into different models.

Traits ^a^	Model G			Model M			Model GM		
	AIC1 ^b^	AIC2	*p*	AIC1	AIC2	*p*	AIC1	AIC2	*p*
BW35	8877.08	8844.91	2.53 × 10^−9^ *	8785.24	8525.86	4.30 × 10^−59^ *	8874.32	8840.77	1.25 × 10^−9^ *
BW70	9552.74	9550.90	2.50 × 10^−2^	9284.30	9198.09	2.95 × 10^−21^ *	9551.21	9549.59	2.86 × 10^−2^
BW84	9852.78	9852.65	7.22 × 10^−2^	9565.13	9487.66	2.45 × 10^−19^ *	9850.87	9851.06	8.94 × 10^−2^
ADG70	4065.08	4064.55	5.59 × 10^−2^	3973.15	3915.19	4.85 × 10^−15^ *	4066.04	4065.44	5.37 × 10^−2^
ADG84	3830.83	3829.54	3.49 × 10^−2^	3740.36	3681.21	2.65 × 10^−15^ *	3831.52	3830.21	3.44 × 10^−2^
FCR70	775.99	770.93	3.96 × 10^−3^ *	771.42	732.82	9.35 × 10^−11^ *	768.73	762.80	2.42 × 10^−3^ *
FCR84	640.19	634.61	2.95 × 10^−3^ *	670.75	617.27	4.72 × 10^−14^ *	639.27	633.58	2.78 × 10^−3^ *
RFI70	5180.47	5182.05	2.58 × 10^−1^	4988.73	4982.32	1.87 × 10^−3^ *	5179.31	5180.84	2.46 × 10^−1^
RFI84	5188.97	5190.02	1.64 × 10^−1^	5015.79	4993.02	3.23 × 10^−3^ *	5190.12	5191.08	1.54 × 10^−1^

^a^ BW35, live body weight at 35 days of age (g); BW70, live body weight at 70 days of age (g); BW84, live body weight at 84 days of age (g); ADG70, average daily gain between 35 and 70 days of age (g/day); ADG84, average daily gain between 35 and 84 days of age (g/day); FCR70, feed conversion ratio between 35 and 70 days of age; FCR84, feed conversion ratio between 35 and 84 days of age; RFI70, residual feed intake between 35 and 70 days of age (g/day); RFI84, residual feed intake between 35 and 84 days of age (g/day). ^b^ AIC1 and AIC2 are the Akaike Information Criterion values for the model including maternal permanent environmental effect or not, respectively; and the *p* values correspond to the likelihood-ratio testing (* *p* < 0.01).

**Table 3 microorganisms-12-02091-t003:** Estimates (±SE) of variance components, heritability, and microbiability for the growth and feed efficiency traits.

Traits ^a^	Models ^b^	Variance Components				Heritability	Microbiability
		$\mathrm{Genetic}\;(\sigma_{a}^{2}$)	$\mathrm{Microbiota}\;(\sigma_{m}^{2}$)	$\mathrm{Maternal}\;(\sigma_{p}^{2}$)	$\mathrm{Residual}\;(\sigma_{e}^{2}$)		
BW35	G	8827	/	7161	3895	0.44 ± 0.13	/
	M	/	588	10,232	7828	/	0.03 ± 0.02
	GM	8566	334	7306	3450	0.44 ± 0.13	0.02 ± 0.01
BW70	G	27,765	/	/	14,491	0.66 ± 0.08	/
	M	/	1178	12,472	26,794	/	0.03 ± 0.03
	GM	27,380	516	/	13,735	0.66 ± 0.08	0.01 ± 0.01
BW84	G	38,656	/	/	22,117	0.64 ± 0.08	/
	M	/	3723	16,470	36,952	/	0.07 ± 0.04
	GM	37,286	2092	/	19,672	0.63 ± 0.08	0.04 ± 0.03
ADG70	G	8.68	/	/	9.51	0.48 ± 0.08	/
	M	/	0.23	3.92	13.47	/	0.01 ± 0.02
	GM	8.59	0.06	/	9.44	0.47 ± 0.08	0.00 ± 0.01
ADG84	G	6.96	/	/	6.83	0.50 ± 0.08	/
	M	/	0.98	3.04	8.89	/	0.08 ± 0.04
	GM	6.30	0.76	/	6.18	0.48 ± 0.09	0.06 ± 0.04
FCR70	G	0.02	/	0.03	0.14	0.11 ± 0.12	/
	M	/	0.01	0.04	0.13	/	0.06 ± 0.04
	GM	0.01	0.00	0.03	0.13	0.06 ± 0.11	0.00 ± 0.03
FCR84	G	0.01	/	0.03	0.11	0.07 ± 0.12	/
	M	/	0.01	0.03	0.11	/	0.07 ± 0.03
	GM	0.01	0.01	0.03	0.10	0.07 ± 0.03	0.07 ± 0.12
RFI70	G	22.58	/	/	78.16	0.22 ± 0.07	/
	M	/	3.55	8.50	85.06	/	0.04 ± 0.03
	GM	19.31	2.69	/	76.00	0.20 ± 0.07	0.03 ± 0.03
RFI84	G	28.80	/	/	68.85	0.29 ± 0.07	/
	M	/	3.04	13.93	79.06	/	0.03 ± 0.03
	GM	27.62	1.18	/	67.60	0.29 ± 0.07	0.01 ± 0.02

^a^ BW35, live body weight at 35 days of age (g); BW70, live body weight at 70 days of age (g); BW84, live body weight at 84 days of age (g); ADG70, average daily gain between 35 and 70 days of age (g/day); ADG84, average daily gain between 35 and 84 days of age (g/day); FCR70, feed conversion ratio between 35 and 70 days of age; FCR84, feed conversion ratio between 35 and 84 days of age; RFI70, residual feed intake between 35 and 70 days of age (g/day); RFI84, residual feed intake between 35 and 84 days of age (g/day). ^b^ G, model G; M, model M; GM, model GM.

**Table 4 microorganisms-12-02091-t004:** Genetic correlations (below diagonal) and standard errors (above diagonal) among the seven heritable traits.

	**BW35**	**BW70**	**BW84**	**ADG70**	**ADG84**	**RFI70**	**RFI84**
**BW35**		0.049	0.054	−0.095	0.106	0.169	0.134
**BW70**	0.762		0.001	0.086	0.096	0.197	0.166
**BW84**	0.708	1.000		0.106	0.059	0.199	0.166
**ADG70**	−0.092	0.579	0.527		0.002	0.221	0.181
**ADG84**	0.023	0.532	0.720	1.000		0.208	0.179
**RFI70**	−0.150	−0.113	−0.067	0.200	0.159		0.076
**RFI84**	−0.061	0.131	0.159	0.192	0.166	0.937	

BW35, live body weight at 35 days of age (g); BW70, live body weight at 70 days of age (g); BW84, live body weight at 84 days of age (g); ADG70, average daily gain between 35 and 70 days of age (g/day); ADG84, average daily gain between 35 and 84 days of age (g/day); RFI70, residual feed intake between 35 and 70 days of age (g/day); RFI84, residual feed intake between 35 and 84 days of age (g/day).

## Data Availability

The data and materials that support the findings of this study can be found in the manuscript. Raw phenotype, genotype, and microbiome data can’t be publicly available as they have been included in our breeding programs.

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
