# Peer review of "Parameter Estimation of Host Genomic and Gut Microbiota Contribution to Growth and Feed Efficiency Traits in Meat Rabbits"

_microorganisms, 2024, doi:10.3390/microorganisms12102091_

Round 1
Reviewer 1 Report
Comments and Suggestions for Authors
The authors investigated the relations between growth and feed efficiency traits, and genomic and gut microbiota data in meat rabbit by means of Statistical modeling. They concluded there were correlations between heritability and all growth traits and one of the two feed efficiency traits, whereas there was no contribution of gut microbiota to phenotypic variation for every trait.
However, the manuscript lacks some information on the concepts and methods of the study.
Comments:
Line 83: Did the authors obtain approval from IACUC? Please provide the ethical statement.
Line 89: I wonder if lighting-assisted estrus synchronization technology is common and necessary in a coprophilic oviparous animal, rabbit.
Line 93: Please explain how the rabbits were finished (euthanized).
Since the only indicator of the animal's condition is its body weight, the weighing condition (time, fasted/fed etc.) should be described in detail.
Wasn’t it possible to measure muscle weight excluding internal organs witch largely affect the body weight, since the authors discuss the productivity of meat.
Line 99: “between 35 and 70 days” should be “between 70 and 84 days”.
Line 106: “for” should be “from”.
Line 116: As mentioned in Line 106.
Comments on the Quality of English LanguageMinor editing of English language required.
Author Response
We really appreciate these suggestions and comments raised by reviewer that greatly contributed to improve our manuscript. We have accordingly revised our manuscript, and in which all changes are highlighted in green. Also, please see the point-by-point responses as below.
Comments 1: The authors investigated the relations between growth and feed efficiency traits, and genomic and gut microbiota data in meat rabbit by means of Statistical modeling. They concluded there were correlations between heritability and all growth traits and one of the two feed efficiency traits, whereas there was no contribution of gut microbiota to phenotypic variation for every trait. However, the manuscript lacks some information on the concepts and methods of the study.
Response 1: We have carefully revised our manuscript, and please see the updated version for details.
Comments 2: Line 83: Did the authors obtain approval from IACUC? Please provide the ethical statement.
Response 2: Thanks for this suggestion. We have added ethical statement in the supplementary section as “The animal study was approved by Institutional Animal Care and Use Committee of Sichuan Agricultural University (2023202046). The study was conducted in accordance with the local legislation and institutional requirements.” (Lines 429-431).
Comments 3: Line 89: I wonder if lighting-assisted estrus synchronization technology is common and necessary in a coprophilic oviparous animal, rabbit.
Response 3: Yes, this technology is commonly used in rabbit industry. To avoid possible confusion, we have added one reference here (doi: 10.1016/j.livsci.2015.11.012; Line 104).
Comments 4: Line 93: Please explain how the rabbits were finished (euthanized).
Response 4: We really appreciate reviewer’s careful reading. We have updated this sentence as “All rabbits were euthanized using an overdose of anesthesia at 84 days of age.” (Line 112).
Comments 5: Since the only indicator of the animal's condition is its body weight, the weighing condition (time, fasted/fed etc.) should be described in detail.
Response 5: The weighting process was re-described more clearly (Lines 112-115).
Comments 6: Wasn’t it possible to measure muscle weight excluding internal organs witch largely affect the body weight, since the authors discuss the productivity of meat.
Response 6: We totally agree that this is a good suggestion. However, we did not include this trait in this study, which may be explored in the future studies.
Comments 7: Line 99: “between 35 and 70 days” should be “between 70 and 84 days”.
Response 7: We are sorry for this mistake. FCR84 is referred to the age interval between 35 and 84 days of age, which was revised accordingly (Line 119).
Comments 8: Line 106: “for” should be “from”.
Response 8: We really appreciate reviewer’s careful reading. This word was updated (Line 127).
Comments 9: Line 116: As mentioned in Line 106.
Response 9: Done (Line 137)
Reviewer 2 Report
Comments and Suggestions for Authors
Line 91: The chemical composition of diets should be available for readers.
Any particular reason why they were offered two different diets?
Were the same diets offered during the whole study?
Line 106: What were the gender of rabbits selected for sampling?
What age of rabbits was the blood sampled?
Line 116: Why were feces samples collected at the 49 days?
Line 132: Birth batch should be a random effect.
Table 1: Why did the number of observations differ between BW, ADG, FCR and RFI?
Is intake data available?
Any idea why their growth rate decreased in d 70-84?
Diet has been shown to significantly impact gut microbiota. For those models showing an animal microbiota effect, how likely is the diet group to impact this result?
Line 204: these results are confusing. ADG is simply calculated from BW between two-time points. How come a strong correlation was observed in BW 70 and 85 but was not in ADG?
Line 207: is it essential to evaluate the BW and ADG correlation? What does this correlation reveal?
In addition, the author stated that r = 0.769, a moderate correlation, but a significant correlation was used to describe BW35 and BW70 at r=0.638. Please define the threshold.
Line 236: Does the maternal permanent environment effect include the gender of progeny?
Line 252: Does the heritability estimate account for the dam and sire?
Line 265: why did the genetic correlation between BW70 and BW84 correspond to ADG70 and ADG84, but BW35 had a strong correlation with BW70 and BW84 but didn’t have a genetic correlation with ADG70 and ADG84?
Line 46. The individual phenotype is a must-have data for genetic selection, whether genome traits are used as a selection index or not.
What is the purpose of finding low inheritable traits?
Line 297: Why did the growth infection occur at 70 years of age? Does living space impact that?
Line 303: The author stated they didn’t have access to pedigree information in Materials and Methods, but here suggested they did.
The tissue used for genomic selection is important for the successful outcome. Why select blood instead of muscle tissue when selecting growth traits?
Can the difference in heritability between study due to the data (pedigree or others) used?
Line 357: Did the time of feces sampled for microbiota have effect on the outcome of correlation?
Samples were collected at only one-time point while BW was measured at d 35, 70, and 84.
Author Response
We really appreciate these suggestions and comments raised by reviewer that greatly contributed to improve our manuscript. We have accordingly revised our manuscript, and in which all changes are highlighted in green. Also, please see the point-by-point responses as below.
Comments 1: Line 91: The chemical composition of diets should be available for readers.
Response 1: Thanks for this suggestion. As we directly used the commercial pellet diets, only a few main nutritional values were labelled. We added this information in the revised manuscript (Lines 106-107).
Comments 2: Any particular reason why they were offered two different diets?
Response 2: Yes, the two source of mineral supplements were used for comparing their effects on individual growth performance. This information was added in the revised manuscript as “The use of different mineral supplements intended to compare the effects on individual growth performances in our other study.” (Lines 107-109).
Comments 3: Were the same diets offered during the whole study?
Response 3: Yes, we used the same diet feeding between 35 and 84 days of age. To avoid possible confusion, we updated this sentence as “Between the weaning at 35 days of age and finishing at 84 days of age, each rabbit …” (Lines 104-105).
Comments 4: Line 106: What were the gender of rabbits selected for sampling? What age of rabbits was the blood sampled?
Response 4: We are sorry for this missing information. We revised this sentence as “At the finishing day, blood samples were collected from 199 rabbits roughly consisting of one male and one female randomly selected per litter” (Lines 127-128).
Comments 5: Line 116: Why were feces samples collected at the 49 days?
Response 5: This is a great question! We added the necessary discussion regarding this issue, as “Furthermore, it is well-known that gut microbiota composition would be dynamically changed along with individual growth. In this study, the 49 days of age of rabbits was selected for sampling the feces samples because of two considerations. First, the gut microbial composition may be relatively stable as there are two weeks after the weaning date. Second, there are at least three weeks remained before the possible finishing date (may be varied from 70 to 84 days of age among different breeds or markets), which could facilitate the obtainment of gut microbiota data for conducting individual selection.” (Lines 403-409).
Comments 6: Line 132: Birth batch should be a random effect.
Response 6: As the lighting-assisted estrus synchronization technology was used, there were only two different birth batches. Therefore, this effect may be preferably analyzed as a fixed effect. We provided the number of levels for each fixed effect in the revised manuscript for better understanding (Lines 153-154).
Comments 7: Table 1: Why did the number of observations differ between BW, ADG, FCR and RFI?
Response 7: Because data quality controls were separately applied to every trait, this may result into different number of observations.
Comments 8: Is intake data available?
Response 8: We don’t have the daily intake records for each rabbit, but the whole intake during each age interval, including between 35 and 70, and between 70 and 84 days of age, were recorded. Based on the intake and weight gain records, we calculated both FCR and RFI for the two age intervals.
Comments 9: Any idea why their growth rate decreased in d 70-84?
Response 9: This suggests that individual growth peak was present around 70 days of age, after where the growth rate would decrease. This is a breed- or population-specific characteristics. Please see our discussion about this finding (Lines 320-322).
Comments 10: Diet has been shown to significantly impact gut microbiota. For those models showing an animal microbiota effect, how likely is the diet group to impact this result?
Response 10: In this study, we did not directly investigate the impact of diet on gut microbiota, which, however, can be commonly found in literature. Furthermore, we did not find the significant contribution of gut microbiota composition to individual growth performance, as being revealed by the estimated microbiabilities. Therefore, the revealed significant effect of diet group on several traits, including BW70, ADG70, ADG84, and RFI70 shown in Table 1, may be mediated by a gut microbiota-independent manner. Certainly, this topic is interesting and needs to be thoroughly explored in future studies.
Comments 11: Line 204: these results are confusing. ADG is simply calculated from BW between two-time points. How come a strong correlation was observed in BW 70 and 85 but was not in ADG?
Response 11: The BW is a measure at a single time point, whereas ADG refers to the BW difference between two time points (further divided by the same number of days). In other words, one rabbit has a greater BW70, but this does not mean that this rabbit still has the greater ADG70 because its BW35 may be also higher. Therefore, different observations on phenotypic correlations can be explained correspondingly.
Comments 12: Line 207: is it essential to evaluate the BW and ADG correlation? What does this correlation reveal?
Response 12: Yes, we think it is necessary to reveal the correlations present between BW and ADG. If strong correlation is present between them, we only need to select either one, otherwise we have to differentially treat them in breeding programs. Although BW and ADG are related by definition, they would indicate different phenotypic characteristics as we explained in Response 11.
Comments 13: In addition, the author stated that r = 0.769, a moderate correlation, but a significant correlation was used to describe BW35 and BW70 at r=0.638. Please define the threshold.
Response 13: We totally understand reviewer’s concern. The correlation degree (strong, moderate, and weak) and statistical significance (the P value was lower than a threshold or not) are two different aspects of a correlation analysis. As we can see in literature, there is no complete consent on the threshold used to strictly classify the strong, moderate, and weak correlations. In this context, we specially provided the estimated r and P values for every correlation results.
Comments 14: Line 236: Does the maternal permanent environment effect include the gender of progeny?
Response 14: Individual gender of progeny and maternal permanent environment are two different effects, and both of them were simultaneously included into the mixed effect model (please see “2.3. Statistical models of single trait”).
Comments 15: Line 252: Does the heritability estimate account for the dam and sire?
Response 15: In this study, we used the single-step genomic BLUP approach for estimating the heritabilities, which integrated all available information together. Therefore, both dam and sire effects (if we understand reviewer’s point correctly) were accounted via the genetic relationships (i.e., the H matrix, Line 170). Additionally, dam’s permanent environment effect was considered.
Comments 16: Line 265: why did the genetic correlation between BW70 and BW84 correspond to ADG70 and ADG84, but BW35 had a strong correlation with BW70 and BW84 but didn’t have a genetic correlation with ADG70 and ADG84?
Response 16: BW35 didn’t have the genetic correlations with ADG70 and ADG84, which means that the magnitude of individual ADG does not depend on the actual BW35. The observed differences of genetic correlations raised by reviewer can be similarly explained by our responses in Response 11.
Comments 17: Line 46. The individual phenotype is a must-have data for genetic selection, whether genome traits are used as a selection index or not.
Response 17: This is true when establishing the reference population of genomic selection. However, individual phenotype is not a must-have data for genetic selection if we have the genomic data available. This is one of the most important advantages regarding genomic selection approaches.
Comments 18: What is the purpose of finding low inheritable traits?
Response 18: For any trait of interest, the first step is to estimate the heritability, which is essential as the magnitude of heritability will help us select the proper selection approaches. For example, we can efficiently select individuals directly based on phenotype and pedigree information for these moderately or highly heritable traits. However, genomic data-based selection may have obvious advantages for traits with low heritability.
Comments 19: Line 297: Why did the growth infection occur at 70 years of age? Does living space impact that?
Response 19: Every species, breed/population, or even each individual would have different growth curve, which is a biological characteristic. Although we do not have experimental data to answer if the living space will impact individual growth curve, we think this factor could be omitted at least in this study as we raised each rabbit per cage individually (Line 105).
Comments 20: Line 303: The author stated they didn’t have access to pedigree information in Materials and Methods, but here suggested they did.
Response 20: We think reviewer confused the two statements. In Materials and Methods (Line 101), we declared that upward pedigree information was not available regarding these parental individuals (revised to “funder individuals”). However, the mating records and pedigree information were carefully collected when crossbreeding these parental individuals to produce this progeny population. Therefore, we subsequently declared to combine genomic and pedigree information to perform the genetic evaluation.
Comments 21: The tissue used for genomic selection is important for the successful outcome. Why select blood instead of muscle tissue when selecting growth traits?
Response 21: As the genomic information is not different for genotyping either tissue or blood samples, we can’t agree that the use of muscle tissue sample is important for the successful outcome of genomic selection. In this study, we used the blood samples only because it is easier for the extraction of genomic DNA.
Comments 22: Can the difference in heritability between study due to the data (pedigree or others) used?
Response 22: Based on the definition of heritability (i.e., the proportion of phenotypic variance explained by the additive genetic effect), its magnitude may be dependent on the specific population. Therefore, the varied estimates of heritability can be explained by any difference in relation to population characteristics, including both genetic (gene frequency, LD degree, genetic relationship matrix constructed, and others) and non-genetic (nutritional and diet supplementation, management practices) factors. Please see our discussion on this issue (Lines 334-336).
Comments 23: Line 357: Did the time of feces sampled for microbiota have effect on the outcome of correlation?
Response 23: Absolutely yes. We added the necessary discussion regarding this issue, as “Furthermore, it is well-known that gut microbiota composition would be dynamically changed along with individual growth. In this study, the 49 days of age of rabbits was selected for sampling the feces samples because of two considerations. First, the gut microbial composition may be relatively stable as there are two weeks after the weaning date. Second, there are at least three weeks remained before the possible finishing date (may be varied from 70 to 84 days of age among different breeds or markets), which could facilitate the obtainment of gut microbiota data for conducting individual selection.” (Lines 403-409).
Comments 24: Samples were collected at only one-time point while BW was measured at d 35, 70, and 84.
Response 24: Yes, we only collected the feces samples only at one time point in this study. The main reason is that we intend to use gut microbiota information to facilitate the individual selection. In this context, the sampled data can only be treated as a molecular marker. Other considerations can be referred to our responses in Response 23.
Reviewer 3 Report
Comments and Suggestions for Authors
Dear authors, to reconsider the article's acceptance for publication, I am providing the following suggestions to improve the manuscript.
Title: I suggest to modifying the title of the paper and other parts of the text to “gut bacteria” or other similar term. Gut microbiota encompasses an entire community of microorganisms that includes bacteria, fungi, viruses, among others.
Abstract: the abstract could highlight the practical implications of the findings for breeders as well the novelty of the study.
Lines 12-13: “Rabbits are a promising livestock to improve global food security due to efficient utilization of plant fibers that are indigestible to human.” What do the authors mean by this?
- Please improve line 24 "did not obtain non-zero estimates of microbiability", is a bit confuse
Introduction:
- This section doesn`t clearly explain the novelty of the study, particularly the integration of genomic and microbiota.
- Please include a definition of “microbiability”
- There are potential factors such as environmental factors that might affect microbiota composition?
- Introduction could be improved with a specific text explaining the role of gut microbiota in the digestion of fiber in herbivorous animals, particularly rabbits, and how this influences feed efficiency and growth. There is a limited understanding of gut microbiota’s role in rabbit growth and feed efficiency?
- Please include a clear statement of the hypothesis of the work
- "Rabbits can efficiently utilize plant fiber fractions that are indigestible to humans" is a valid, but this can be expanded to explain why this makes rabbits a suitable model?
MMs:
- Why New Zealand White rabbits specifically? findings can be extended to other breeds or species?
- The study uses 16S rRNA gene sequencing for gut microbiota. Authors think that this is a limitation because this method only captures bacterial composition and no other microbial domains?
- The microbial analysis did not yield significant results; this raises concerns for the authors about the approach? Microbial BLUP (MBLUP) to estimate microbiota is a method that cannot capture non-linear relationships between microbial species and their traits. This can affect the host-microbiota interactions interpretations?
- The paper excluded low-abundance ASVs (amplicon sequence variants) that appeared in fewer than 20% of samples. However, low-abundance species might play crucial roles in regulating host traits. Authors considered that more sensitivity analyses could assess the impact of these exclusions on the results?
- Environmental factors were controlled?
- Was the work evaluated by the ethics committee on the use of animals in science before it began?
Results and discussion:
- The FCR was found to have low heritability, while RFI had moderate heritability (0.29 for RFI84). The authors suggest that RFI may be a better trait for genetic improvement of feed efficiency in rabbits. However, this contrasts with the findings in other species, where both FCR and RFI are moderately heritable. How authors can explain this? Please provide comparable heritability estimates for BW and feed efficiency traits in rabbits or other species.
- The study indicated that no significant microbiability was detected for any of the growth and feed efficiency parameters. This contrasts with other studies. However, alternative explanations, such as insufficient sequencing depth, variability in microbiota data, or the use of inappropriate models should also be considered. Authors could include this and suggest future studies in this field. Please make a more detailed discussion of the microbiota findings, why the results contrast with previous studies?
- The paper suggests that the gut microbiota did not contribute significantly contrasting with findings in other livestock. This discrepancy might be due to sample size limitations? only 199 out of 739 rabbits genotyped? And the breed effect? This partial genotyping could reduce the accuracy of the genetic estimates?
- The reported genetic correlation of 1.000 between BW70 and BW84 is unusually high. The paper could discuss why such a strong correlation exists. Especially considering the perfect correlation between BW70 and BW84.
- Information explaining that a more rigorous control of environmental factors such as housing conditions and diet composition would reduce variables that could impact the gut microbiota and trait correlations.
- There are potential confounders that may affect gut microbiota composition and growth traits, such as diet, housing conditions, and environmental variables? microbiota change over time?
- Please include in discussion the potential biases introduced due to the small genotyping sample in relation to the total population.
- The figure descriptions for phenotypic correlations (Figure 1) are accurate, but these correlations were expected? how it can be compared to other rabbit or livestock species?
- Finally, how these findings can influence future breeding strategies or practical applications for improving rabbit production?
Conclusion: this section doesn`t address the potential limitations of the study (e.g., small sample size for genotyping, exclusion of low-abundance ASVs). Please clarify the limitations of the study, particularly in terms of sample size, genotyping, and microbiota analysis. In addition, expanding on the practical implications of the study for breeding programs such as implications of heritability as well breeding strategies could improve the conclusion. And the future research directions?
Comments on the Quality of English LanguageMinor editing
Author Response
We really appreciate these suggestions and comments raised by reviewer that greatly contributed to improve our manuscript. We have accordingly revised our manuscript, and in which all changes are highlighted in green. Also, please see the point-by-point responses as below.
Comments 1: Dear authors, to reconsider the article's acceptance for publication, I am providing the following suggestions to improve the manuscript.
Response 1: According to these suggestions and comments, we have carefully revised our manuscript.
Comments 2: Title: I suggest to modifying the title of the paper and other parts of the text to “gut bacteria” or other similar term. Gut microbiota encompasses an entire community of microorganisms that includes bacteria, fungi, viruses, among others.
Response 2: We totally understand reviewer’s concern. Strictly speaking, gut microbiota is the collection of bacteria, archaea, fungi, and viruses. However, bacteria make up the vast majority of the gut microbiota. Therefore, there are a large number of publications that have used the term of “gut microbiota” to describe gut bacterial composition. Here, we would like to keep consistent with these publications, and we think this will not confuse readers in this field.
Comments 3: Abstract: the abstract could highlight the practical implications of the findings for breeders as well the novelty of the study.
Response 3: Thanks for this suggestion. We have updated the related sentence in the revised manuscript as “Based on the combined use of genomic and gut microbiota data, the parameters obtained in this study could help us to implement efficient breeding schemes in meat rabbits” (Lines 26-28).
Comments 4: Lines 12-13: “Rabbits are a promising livestock to improve global food security due to efficient utilization of plant fibers that are indigestible to human.” What do the authors mean by this?
Response 4: We are sorry for this confusing sentence, which was rephrased as “… may contribute to the alleviation of feed-food competition” (Lines 12-13, 38).
Comments 5: - Please improve line 24 "did not obtain non-zero estimates of microbiability", is a bit confuse
Response 5: This sentence was rephrased as “… whereas the estimates did not significantly deviate from zero for the two FCR traits” (Lines 22-23).
Comments 6: Introduction:
- This section doesn’t clearly explain the novelty of the study, particularly the integration of genomic and microbiota.
Response 6: Thanks for this helpful suggestion. We have additionally described the advantages about combining genomic and gut microbiota data together. Please see the revised manuscript for details (Lines 77-80).
Comments 7: - Please include a definition of “microbiability”
Response 7: We added the definition of microbiability (Lines 68-70).
Comments 8: - There are potential factors such as environmental factors that might affect microbiota composition?
Response 8: Yes, there are many environmental factors that will impact the gut microbiota composition. We addressed this issue in the revised manuscript (Lines 81-82).
Comments 9: - Introduction could be improved with a specific text explaining the role of gut microbiota in the digestion of fiber in herbivorous animals, particularly rabbits, and how this influences feed efficiency and growth. There is a limited understanding of gut microbiota’s role in rabbit growth and feed efficiency?
Response 9: We really appreciate this suggestion. We addressed this issue in the revised manuscript as “In ruminants, gastrointestinal microbiota has been extensively involved in the digestion of recalcitrant dietary fiber [12], which can further impact host growth performance and feed efficiency [13,14]. However, the impacts of gut microbiota on the digestion of dietary fiber, growth, and feed efficiency have not been thoroughly explored yet in rabbits [10,15]” (Lines 57-61).
Comments 10: - Please include a clear statement of the hypothesis of the work
Response 10: We provided the clear statement of the hypothesis in the revised manuscript as “Under the hypothesis that host genetic and/or gut microbiota impact the growth and freed efficiency in rabbits, therefore, the main objectives …” (Lines 91-92).
Comments 11: - "Rabbits can efficiently utilize plant fiber fractions that are indigestible to humans" is a valid, but this can be expanded to explain why this makes rabbits a suitable model?
Response 11: The main objectives of this study are to estimate the related parameters accounting for host genomic and gut microbiota contribution to growth and feed efficiency traits in meat rabbits. As these parameters are breed/population-dependent, we can’t treat this population, breed, or species as any suitable model.
Comments 12: MMs:
- Why New Zealand White rabbits specifically? findings can be extended to other breeds or species?
Response 12: In this study, we selected New Zealand White rabbits mainly because it is an excellent breed of meat rabbits raised in our research farm, and this information was added (Line 100). As we discussed in Response 11, these findings would not directly be extended to other breeds or species. As you may find in the Discussion section (Lines 326-336), these parameters estimated in this study are some different from that reported in literature.
Comments 13: - The study uses 16S rRNA gene sequencing for gut microbiota. Authors think that this is a limitation because this method only captures bacterial composition and no other microbial domains?
Response 13: We can’t absolutely exclude the possibility that other microbial domains may have impacts on individual growth and feed efficiency. However, as discussed in Response 2, we think that the profiling of bacterial composition is enough to achieve our research objectives included in this study.
Comments 14: - The microbial analysis did not yield significant results; this raises concerns for the authors about the approach? Microbial BLUP (MBLUP) to estimate microbiota is a method that cannot capture non-linear relationships between microbial species and their traits. This can affect the host-microbiota interactions interpretations?
Response 14: We totally agree with reviewer. Similarly, we can’t exclude the possibility that there would be non-linear relationships between microbial composition and individual phenotypic traits, which can be further explored from the methodological point of view, such as using non-linear regression and machine learning approaches. In this study, we employed BLUP approach as it has been most widely used for genetic evaluation in both animals and plants. Furthermore, greater sample size may be required for accounting for the non-linear relationships.
Comments 15: - The paper excluded low-abundance ASVs (amplicon sequence variants) that appeared in fewer than 20% of samples. However, low-abundance species might play crucial roles in regulating host traits. Authors considered that more sensitivity analyses could assess the impact of these exclusions on the results?
Response 15: In this study, both growth and feed efficiency are the continuous traits and may have the polygenic and polymicrobial regulation bases. In theory, these sparse ASVs may have the limited contributions to these continuous traits. Second, the inclusion of too many sparse ASVs will more likely make the relationship matrix not positive definite. Therefore, it is a widely used strategy in literature to remove these sparse ASVs in prior to the calculation of relationship matrix.
Comments 16: - Environmental factors were controlled?
Response 16: In this study, all rabbits had been raised in the same barn, and with the same management practices. Other known non-genetic factors, including gender, birth batch, and diet group, were specifically accounted into the mixed effect model to properly adjust their effects. Please see the two sections of “2.1. Animals and phenotypes” and “2.3. Statistical models of single trait” for details.
Comments 17: - Was the work evaluated by the ethics committee on the use of animals in science before it began?
Response 17: Thanks for this suggestion. We have added the ethical statement in the supplementary section as “The animal study was approved by Institutional Animal Care and Use Committee of Sichuan Agricultural University (2023202046). The study was conducted in accordance with the local legislation and institutional requirements.” (Lines 429-431).
Comments 18: Results and discussion:
- The FCR was found to have low heritability, while RFI had moderate heritability (0.29 for RFI84). The authors suggest that RFI may be a better trait for genetic improvement of feed efficiency in rabbits. However, this contrasts with the findings in other species, where both FCR and RFI are moderately heritable. How authors can explain this? Please provide comparable heritability estimates for BW and feed efficiency traits in rabbits or other species.
Response 18: Thanks for this helpful suggestion. Considering the definition of heritability, the varied estimates among different studies may be essentially resulted from the inter-population differences on both genetic (gene frequency, LD degree, genetic relationship matrix constructed, and others) and non-genetic (nutritional and diet supplementation, management practices) factors. We added the related discussion in the revised manuscript (Lines 334-336). The similar publications reporting the estimates of heritability in rabbits have been cited in this manuscript, such as reference 6, 56, 57, and 58.
Comments 19: - The study indicated that no significant microbiability was detected for any of the growth and feed efficiency parameters. This contrasts with other studies. However, alternative explanations, such as insufficient sequencing depth, variability in microbiota data, or the use of inappropriate models should also be considered. Authors could include this and suggest future studies in this field. Please make a more detailed discussion of the microbiota findings, why the results contrast with previous studies?
Response 19: We agree with reviewer on this point, which were specially addressed. Please see the revised manuscript for details (Lines 387-391).
Comments 20: - The paper suggests that the gut microbiota did not contribute significantly contrasting with findings in other livestock. This discrepancy might be due to sample size limitations? only 199 out of 739 rabbits genotyped? And the breed effect? This partial genotyping could reduce the accuracy of the genetic estimates?
Response 20: First of all, the number of animals genotyped doesn’t influence the microbial results as we actually obtained the comparable estimates when didn’t include genotype data into the mixed effect model (i.e., the model M). In the context of genomic evaluation, the estimation accuracy of genetic parameters, as well as of individual breeding values, may be improved more or less if all individuals are genotyped. However, it is necessary to balance the cost and benefits in any breeding program, especially for rabbits and other small livestock. Because of the precise recordings of individual mating in this population, therefore, only about 30% of individuals were genotyped in this study and then subjected to single-step GBLUP approach. Furthermore, the required sample size may be decreased as the raising environments are homogenous for all rabbits. Collectively, we believe that the sample size is not a critical factor influencing the estimated heritabilities, which can actually be indicated by the relatively small values of SE. As we discussed in Response 19, different breeds, even different populations of same breed, are expected to have the contrasting estimates of genetic and microbial parameters.
Comments 21: - The reported genetic correlation of 1.000 between BW70 and BW84 is unusually high. The paper could discuss why such a strong correlation exists. Especially considering the perfect correlation between BW70 and BW84.
Response 21: We only know that a perfect genetic correlation indicates the two traits are completely identical on the genetic basis. Accordingly, we added one sentence to indicate this issue as “For BW, ADG and RFI, strong phenotypic and nearly complete genetic correlations were observed between 70 and 84 days of age, which means that the two traits could be replaceable with each other into breeding schemes. For example, only either of BW70 and BW84 needs to be included for selection” (Lines 365-368).
Comments 22: - Information explaining that a more rigorous control of environmental factors such as housing conditions and diet composition would reduce variables that could impact the gut microbiota and trait correlations.
Response 22: Thanks for this suggestion. These issues have been addressed, please see the section of “2.1. Animals and phenotypes” in the revised manuscript.
Comments 23: - There are potential confounders that may affect gut microbiota composition and growth traits, such as diet, housing conditions, and environmental variables? microbiota change over time?
Response 23: As you can find in the section of “2.1. Animals and phenotypes”, the possible environmental variables were controlled as well as possible. Other known factors, such as birth batch and diet group, were further included in the models for properly adjusting their effects. Of course, we can’t absolutely exclude all potential environmental variables, which however will enter into the residual parts. Absolutely yes, gut microbiota composition must change over time because it is dynamic. Regarding such dynamic profiling, as well as the gene expression levels, it is still a controversy issue about when to collect and analyze the biological samples. In this study, we collected feces samples at the 49 days of age, and the reasons were additionally discussed in the revised manuscript (Lines 403-409).
Comments 24: - Please include in discussion the potential biases introduced due to the small genotyping sample in relation to the total population.
Response 24: According to this suggestion, we provided the necessary discussion for this issue. Please see the revised manuscript (Lines 347-349).
Comments 25: - The figure descriptions for phenotypic correlations (Figure 1) are accurate, but these correlations were expected? how it can be compared to other rabbit or livestock species?
Response 25: Actually, we don’t have any expectation on these correlations, as they may be population-dependent. What we can do is to present, explain, and further utilize these estimation values when designing our breeding programs. In meat rabbits, very few literatures are available for systematically comparing these parameters across different studies, which, however, could be analyzed using the meta-analysis approaches in future studies. Thanks for this comment!
Comments 26: - Finally, how these findings can influence future breeding strategies or practical applications for improving rabbit production?
Response 26: We highlighted this point in the Conclusion section, as indicated by the next Response 27.
Comments 27: Conclusion: this section doesn’t address the potential limitations of the study (e.g., small sample size for genotyping, exclusion of low-abundance ASVs). Please clarify the limitations of the study, particularly in terms of sample size, genotyping, and microbiota analysis. In addition, expanding on the practical implications of the study for breeding programs such as implications of heritability as well breeding strategies could improve the conclusion. And the future research directions?
Response 27: We greatly appreciate these suggestions. We have rephrased the Conclusion, and please see the revised manuscript for details (Lines 411-421).
Round 2
Reviewer 3 Report
Comments and Suggestions for Authors
The authors responded to my suggestions and they have addressed all the comments appropriately. In my opinion, the manuscript is now ready for acceptance.